

# Conformal field theory for inhomogeneous one-dimensional quantum systems: the example of non-interacting Fermi gases

**Jérôme Dubail[1], Jean-Marie Stéphan[2,3*], Jacopo Viti[4] and Pasquale Calabrese[5]**

**1** CNRS & IJL-UMR 7198, Université de Lorraine, F-54506 Vandoeuvre-les-Nancy, France
**2** Max Planck Institut für Physik komplexer Systeme, Nöthnitzer Str. 38 D01187 Dresden, Germany
**3** CNRS & Institut Camille Jordan-UMR 5208, Université Lyon 1, 69622 Villeurbanne, France
**4** ECT & Instituto Internacional de Fisica, UFRN, Lagoa Nova 59078-970 Natal, Brazil
**5** SISSA and INFN, Via Bonomea 265, 34136 Trieste, Italy

\* stephan@math.univ-lyon1.fr

## Abstract

**Conformal field theory (CFT) has been extremely successful in describing large-scale universal effects in one-dimensional (1D) systems at quantum critical points. Unfortunately, its applicability in condensed matter physics has been limited to situations in which the bulk is uniform because CFT describes low-energy excitations around some energy scale, taken to be constant throughout the system. However, in many experimental contexts, such as quantum gases in trapping potentials and in several out-of-equilibrium situations, systems are strongly inhomogeneous. We show here that the powerful CFT methods can be extended to deal with such 1D situations, providing a few concrete examples for non-interacting Fermi gases. The system's inhomogeneity enters the field theory action through parameters that vary with position; in particular, the metric itself varies, resulting in a CFT in curved space. This approach allows us to derive exact formulas for entanglement entropies which were not known by other means.**



## Contents



# 1   Introduction

Low-dimensional quantum systems are a formidable arena for the study of many-body physics: in one or two spatial dimensions (1D or 2D), the effects of strong correlations and interactions are enhanced and lead to dramatic effects. Celebrated examples from condensed matter physics include such diverse cases as the fractionalization of charge and emergence of topological order in the quantum Hall effect, high-$T_c$ superconductivity, or the breakdown of Landau's Fermi liquid theory in 1D, replaced by the Luttinger liquid paradigm [1]. In the past decade, breakthroughs in the field of optically trapped ultra-cold atomic gases [2] have lead to a new generation of quantum experiments that allow to directly observe fundamental phenomena such as quantum phase transitions [3] and coherent quantum dynamics [4,5] in low-dimensional systems, including 1D gases [6–11]. These revolutionary experiments are an ideal playground for the interplay with theory, as they allow to directly realize, in the laboratory, ideal setups that were previously regarded only as oversimplified thought experiments.

On the theory side, many exact results in 1D can be obtained by a blend of methods that comprises lattice integrability [12,13] and non-perturbative field theory approaches, in particular 2D (1+1D) conformal field theory (CFT) [14,15] and integrable field theory [16]. CFT has been incredibly successful at making exact universal predictions for 1D condensed matter systems at a quantum critical point; these include the Kondo effect and other quantum impurity problems [17], and the many insights on quantum quenches [18] as well as universal characterization of entanglement at quantum criticality [19–21]. Entanglement entropies, in particular, are currently in the limelight, as they have become experimentally measurable both in- [22] and out-of-equilibrium [23], opening the route to a direct comparison between experimental data and many exact analytical results obtained by CFT. Indeed, entanglement entropies are usually difficult to compute in microscopic models [24, 25], but their scaling limit is obtained within the powerful CFT approach by solving elementary exercises on Riemann surfaces [21].

There is a caveat in the CFT approach to 1D physics though: since it describes low-energy excitations around some fixed energy scale (*e.g.* the Fermi energy), CFT does not accommodate strong variations of that scale throughout the system. This rules out *a priori* the possibility of tackling *inhomogeneous* systems, in which the relevant energy scale varies. This caveat is important, as inhomogeneous systems are the rule rather than the exception in the realm of quantum experiments: quantum gases at equilibrium always lie in trapping potentials (often

harmonic) and therefore usually come with a non-uniform density; this is also the case of many out-of-equilibrium situations, such as trap releases.

In this paper, we make one step forward. We focus on the example of the free Fermi gas, in a few illustrative in- and out-of-equilibrium inhomogeneous situations. The free Fermi gas is technically simpler than interacting models, and yet it allows to draw interesting lessons that will hold more generally. We find that the varying energy scale is taken into account rather naturally in the effective field theory, in the form of varying parameters in the action. Interestingly, the *metric* is one such parameter, so one generically ends up with a CFT in curved 2D space. These conclusions are very general and hold under the reasonable assumption of separation of scales: there must exist an intermediate scale $\ell$ which is simultaneously large compared to the microscopic scale (the inter-particle distance $\sim \rho^{-1}$, where $\rho$ is the density), but small compared to the scale on which physical quantities vary macroscopically (of order $\rho|\partial_x \rho|^{-1}$). Indeed, at the intermediate scale $\ell$, the system is well described by continuous fields because $\rho^{-1} \ll \ell$, and is locally homogeneous because $\ell|\partial_x \rho|/\rho \ll 1$, so one knows that it corresponds to a standard (i.e. flat-space, translation-invariant) field theory. From there, it is clear that unravelling the global theory for the *inhomogeneous* system is a problem of geometric nature: it is about understanding the global geometric data (*e.g.* metric tensor, coupling constants, gauge fields, *etc.*) that enter the action. This is the program we illustrate with the few simple examples below. We demonstrate the power of this formalism by providing new exact asymptotic formulas for entanglement entropies.

## 2 The one-dimensional Fermi gas

### 2.1 Translation-invariant Fermi gas and the euclidean 2D Dirac action

Let us start by considering a free Fermi gas in 1D

$$H = \int_{-\infty}^{\infty} dx \, c^{\dagger}(x) \left[ -\frac{\hbar^2}{2m} \partial_x^2 - \mu + V(x) \right] c(x), \tag{1}$$

in the absence of an external potential, i.e. $V(x) = 0$. For the reader's convenience, we briefly review the well known relation between this (homogeneous) 1D system and the CFT of massless Dirac fermions in 2D euclidean space-time. The question is: *what is the proper field theory framework that captures the behavior of long-range correlations of arbitrary local observables* $\langle \phi_1(x_1, y_1) \dots \phi_n(x_n, y_n) \rangle$? Here and below, the $y$-coordinate is imaginary time. The starting point to answer this question is the ground-state propagator

$$\left\langle c^{\dagger}(x, y) c(0, 0) \right\rangle = \int_{-k_{\mathrm{F}}}^{k_{\mathrm{F}}} \frac{dk}{2\pi} e^{-i\left[ kx + i\varepsilon(k)\frac{y}{\hbar} \right]}, \tag{2}$$

where $\varepsilon(k) = \frac{\hbar^2 k^2}{2m} - \mu$ is the dispersion relation and $k_F = \frac{1}{\hbar}\sqrt{2m\mu}$ is the Fermi momentum. Its large distance behavior is obtained by linearizing the dispersion relation around the two Fermi points $k_{\mathrm{F}}^{\pm} = \pm k_F$, $\varepsilon(k) \simeq \pm v_{\mathrm{F}} \hbar (k \mp k_{\mathrm{F}})$ with Fermi velocity $v_{\mathrm{F}} = \frac{1}{\hbar} \frac{d\varepsilon}{dk}|_{k=k_F}$. One finds straightforwardly, for $x, v_{\mathrm{F}} y \gg 1/k_{\mathrm{F}}$,

$$
\begin{aligned}
\left\langle c^{\dagger}(x, y) c(0, 0) \right\rangle &\simeq \int_{-\infty}^{k_F} \frac{dk}{2\pi} e^{-i[kx + i(k - k_{\mathrm{F}})v_{\mathrm{F}} y]} + \int_{-k_F}^{\infty} \frac{dk}{2\pi} e^{-i[kx - i(k + k_{\mathrm{F}})v_{\mathrm{F}} y]} \\
&= \frac{i}{2\pi} \left[ \frac{e^{-ik_{\mathrm{F}}x}}{x + iv_{\mathrm{F}}y} - \frac{e^{ik_{\mathrm{F}}x}}{x - iv_{\mathrm{F}}y} \right].
\end{aligned}
\tag{3}
$$

These two terms coincide with the right(R)-/left(L)-components of a massless Dirac fermion in 2D euclidean spacetime, $\langle \psi_{\mathrm{R,L}}^\dagger(x,y)\psi_{\mathrm{R,L}}(0,0)\rangle = \frac{1}{x\pm i v_F y}$, that derive from the action (with $z = x + i v_{\mathrm{F}} y$, $\bar{z} = x - i v_{\mathrm{F}} y$):

$$\mathscr{S} = \frac{1}{\pi} \int dz d\bar{z} \left[ \psi_{\mathrm{R}}^\dagger \partial_{\bar{z}} \psi_{\mathrm{R}} + \psi_{\mathrm{L}}^\dagger \partial_z \psi_{\mathrm{L}} \right]. \tag{4}$$

The action (4) is invariant under conformal transformations $z \mapsto f(z)$, $\psi_{\mathrm{R}}(z) \mapsto (\frac{df}{dz})^{\frac{1}{2}} \psi_{\mathrm{R}}(f(z))$, $\psi_L(\bar{z}) \mapsto (\frac{d\bar{f}}{d\bar{z}})^{\frac{1}{2}} \psi_R(\bar{f}(\bar{z}))$, where $f(z)$ is a holomorphic function of $z$. The phase factors in (3) may be incorporated into (4) with a chiral gauge transformation $\psi_{\mathrm{R,L}} \to e^{\pm i\alpha(x)}\psi_{\mathrm{R,L}}$. Here we do not keep track of these phase factors, as they are unimportant for our purposes, and simply discard them; these aspects are discussed in appendix A.

From Wick's theorem, it then follows that the large-scale behavior of arbitrary multi-point correlations of the original fermions can be obtained from correlators in the massless Dirac theory.

## 2.2 Harmonic trap and the euclidean Dirac action in curved 2d space

Consider the Fermi gas (1) in a harmonic potential $V(x) = m\omega^2 x^2/2$. We ask: *what is the underlying Dirac action*? The system is now *inhomogeneous*, so there should be parameters in the effective action that vary with position. But *what are these parameters* in (4)? In order to understand this, one needs to find the density profile first. The latter is obtained from the exact solution of the microscopic problem in the thermodynamic limit, which in this case is extremely simple. The single-particle eigenstates are just those of the harmonic oscillator, and the many-body ground state is obtained by simply filling up all eigenstates with negative energies. In the thermodynamic limit ($\mu \gg \hbar\omega$), the density profile $\rho(x) = \langle c^\dagger(x)c(x)\rangle$ follows the Wigner semicircle law

$$\rho(x) = \frac{1}{\pi} \sqrt{2(\mu - x^2/2)}, \tag{5}$$

which is non-vanishing in the interval $[-L, L]$ with $L = \sqrt{2\mu}$. Here and in the following, we set $\hbar = m = \omega = 1$. The total number of particles is $N = \int \rho(x)dx = \mu \gg 1$. Away from the edges $x = \pm L$, there is an intermediate scale $\ell$ such that $N^{-\frac{1}{2}} \sim \rho^{-1} \ll \ell \ll L \sim N^{\frac{1}{2}}$. At this scale, the system can be viewed as homogeneous, with a local Fermi momentum $k_{\mathrm{F}}(x) = \pi\rho(x)$: in a window of width $\sim \ell$ around the position $x$, the system consists of all states filled in the interval $[-k_{\mathrm{F}}(x), +k_{\mathrm{F}}(x)]$. Thus, around a point $(x, y) \in [-L, L] \times \mathbb{R}$ in spacetime, the behaviour of the propagator must be the same as in the translation-invariant gas with $k_{\mathrm{F}} = k_{\mathrm{F}}(x)$:

$$\left\langle c^\dagger(x + \delta x, y + \delta y)c(x, y)\right\rangle \simeq \frac{i}{2\pi}\left[ \frac{e^{-i\delta\varphi(x,y)}}{\delta x + i v_{\mathrm{F}}(x)\delta y} - \frac{e^{i\delta\overline{\varphi}(x,y)}}{\delta x - i v_{\mathrm{F}}(x)\delta y} \right], \tag{6}$$

where $v_{\mathrm{F}}(x) = \varepsilon'(k_{\mathrm{F}}(x))$, $\delta\varphi = k_F(x)\delta x + i v_F(k_F(x))\delta y$, and $\delta\overline{\varphi}$ is its complex conjugate. Like in (3), we would like to view the terms $(\delta x \pm i v_{\mathrm{F}}(x)\delta y)^{-1}$ as the R-/L-components of a massless Dirac field. Of course, in the neighborhood of $(x, y)$, one can always do that. But the real question is: *is there a consistent Dirac theory defined globally on the entire domain $(x, y) \in [-L, L] \times \mathbb{R}$, such that its propagator has the required local behavior everywhere*?

If the reader is familiar with quantum field theories in curved background, they will probably have guessed that Eq. (6) is, in fact, related to the propagator of the massless Dirac fermion *in curved space*. The action of the latter theory in 2D is

$$\mathscr{S} = \frac{1}{2\pi} \int dz d\bar{z} \, e^{\sigma(x,y)} \left[ \psi_{\mathrm{R}}^\dagger \overleftrightarrow{\partial}_{\bar{z}} \psi_{\mathrm{R}} + \psi_{\mathrm{L}}^\dagger \overleftrightarrow{\partial}_z \psi_{\mathrm{L}} \right], \tag{7}$$

written in isothermal coordinates $(z, \bar{z})$ and in a fixed frame (see appendix A for all details). The underlying Riemannian metric is

$$ds^2 = e^{2\sigma} dz d\bar{z}. \tag{8}$$

To connect this theory to Eq. (6), notice that, in the coordinate $z$, the propagator behaves locally as

$$\langle \psi_R^\dagger(z + \delta z)\psi_R(z)\rangle = \frac{1}{e^\sigma \delta z}. \tag{9}$$

Thus, to prove that Eq. (6) is the propagator of a Dirac fermion in a curved metric, it is sufficient to exhibit a map $(x, y) \mapsto z(x, y)$ such that

$$e^{\sigma(x,y)} \delta z(x, y) = \delta x + i v_F(x) \delta y, \tag{10}$$

for some real-valued function $\sigma(x, y)$. This equation can be solved easily. First, notice that it is equivalent to $e^\sigma \partial_x z = 1$, $e^\sigma \partial_y z = i v_F$. Writing that $\partial_x \partial_y z = \partial_y \partial_x z$, we find a constraint on $\sigma$: $(i v_F \partial_x - \partial_y)\sigma = i \partial_x v_F$. Looking for a solution that is independent of $y$, we can set $e^{\sigma(x)} = v_F(x)$, which implies $\partial_x z = 1/v_F(x)$ and $\partial_y z = 1$. The solution is straightforward: up to an additive constant, we find the complex coordinate system $(z, \bar{z})$, which lives on the infinite strip $[-\frac{\pi}{2}, \frac{\pi}{2}] \times \mathbb{R}$:

$$z(x, y) = \arcsin\left(\frac{x}{L}\right) + i y, \quad e^\sigma = v_F = \sqrt{L^2 - x^2}. \tag{11}$$

This fixes the underlying geometry of the problem.

# 3 Application to the entanglement entropy

To illustrate the power of this approach, we exhibit new exact results for the entanglement entropies of the Fermi gas in external trapping potentials. Such calculations are, in general, difficult. But within the framework we just developed, they boil down to elementary manipulations of complex analytic functions. The Renyi entanglement entropies of a subsystem $A$ are defined as

$$S_n = \frac{1}{1-n} \ln \mathrm{tr}(\rho_A^n), \tag{12}$$

where $\rho_A$ is the reduced density matrix of $A$ and $n$ is an arbitrary real number. For $n \to 1$, $S_n$ reduces to the von Neumann entropy of the subsystem which is usually referred to as entanglement entropy.

The main property we will use in the following is that the Renyi entanglement entropies are related to the expectation values of the twist fields $\mathcal{T}_n$ [21, 26, 27] which under conformal mapping share the same transformation properties of primaries with dimension

$$\Delta_n = \frac{c}{12}\left(n - \frac{1}{n}\right), \tag{13}$$

with $c$ the central charge of the CFT ($c = 1$ for the free Fermi gas).

## 3.1 Entanglement entropy of the Fermi gas in a harmonic trap

Let us now apply Eqs. (8)-(11) to the problem of calculating the entanglement entropy in a harmonic trap.

We start with the case of a bipartition $A \cup B$ consisting of two semi-infinite systems, $A = [-\infty, x]$, $B = [x, +\infty]$. In a homogeneous system, the Renyi entanglement entropy is [21]

$$S_n(x) \simeq \frac{1}{1-n} \ln \epsilon^{\Delta_n} \langle \mathscr{T}_n(x, y=0) \rangle. \tag{14}$$

$\epsilon$ is a UV cutoff, sometimes dismissed in homogeneous systems, because it simply appears in the form of a non-universal constant offset. In *inhomogeneous* situations, however, it is crucial to have a closer look at this cutoff: since the energy scale changes throughout the system, why shouldn't $\epsilon$ vary as well? And, if $\epsilon(x)$ varies, then it must obviously affect the dependence of $S_n$ on $x$. And indeed, there is a good reason why $\epsilon$ should vary with position: the continuous Fermi gas is locally galilean invariant, and the only relevant microscopic scale is the inverse density $1/\rho(x)$, or equivalently $k_F^{-1}(x)$. So the UV cutoff must simply be proportional to that scale: $\epsilon(x) = \epsilon_0/k_F(x)$, for some dimensionless constant $\epsilon_0$.

Coming back to the harmonic potential, we now make use of the coordinate system $(z, \bar{z})$, with $z(x, y) = \arcsin(x/L) + iy$. To evaluate $\langle \mathscr{T}_n \rangle$, we first perform a Weyl transformation $e^{2\sigma} dz d\bar{z} \to dz d\bar{z}$, which changes $\langle \mathscr{T}_n \rangle$ into $e^{\sigma \Delta_n} \langle \mathscr{T}_n \rangle$. Next, we notice that under the $z \mapsto g(z)$ mapping defined above, $\langle \mathscr{T}_n(z, \bar{z}) \rangle$ becomes $\left| \frac{dg(z)}{dz} \right|^{\Delta_n} \langle \mathscr{T}_n(g(z), \overline{g}(\bar{z})) \rangle_{\text{uhp}}$. The latter factor, which is the one-point function in the upper half-plane, is equal to $(\operatorname{Im} g(z))^{-\Delta_n}$. Putting everything together,

$$\langle \mathscr{T}_n(z, \overline{z}) \rangle = \left( e^{\sigma(z, \bar{z})} \left| \frac{dg(z)}{dz} \right|^{-1} \operatorname{Im} g(z) \right)^{-\Delta_n}. \tag{15}$$

Hence, using Eq. (14) with $\epsilon(x) = \epsilon_0/k_F(x)$, we finally have for the entanglement entropy

$$S_n = \frac{n+1}{12n} \ln \left[ k_F(x) e^{\sigma(z, \bar{z})} \left| \frac{dg(z)}{dz} \right|^{-1} \operatorname{Im} g(z) \right], \tag{16}$$

up to an additive constant and subleading corrections, which we systematically drop from now on. This gives

$$S_n(x) = \frac{n+1}{12n} \ln \left[ L^2 \left( 1 - (x/L)^2 \right)^{3/2} \right]. \tag{17}$$

A more complicated bipartition $A \cup B$ that can be considered in our framework is $A = [x_1, x_2]$ and $B = [-\infty, x_1] \cup [x_2, +\infty]$, where $-L < x_1 < x_2 < L$. The calculation is straightforward, but rather cumbersome and so we report it in appendix B.1. The final result, setting $\zeta_i = x_i/L$, can be written as

$$S_n = \frac{n+1}{6n} \left[ \ln L^2 + \ln \frac{(1-\zeta_1^2)^{3/4}(1-\zeta_2^2)^{3/4}}{2(\zeta_2 - \zeta_1)} + \ln \left( 1 - \zeta_1 \zeta_2 - \sqrt{(1-\zeta_1^2)(1-\zeta_2^2)} \right) \right], \tag{18}$$

which is a highly non-trivial generalization of recent results (for $x_2 = -x_1$) obtained by means of random matrix theory [28]. We checked the validity of this formula against exact finite size computations for lattice and continuous Fermi gases using the approaches of Refs. [29,30].

## 3.2 Fermi gas in an arbitrary external potential

The generalization to arbitrary $V(x)$ is very simple, even though the single particle problem is not always exactly solvable for a general potential. Indeed, we are always interested in the thermodynamic limit, where the single particle states that matter are those up in the spectrum, and for which the semi-classical approximation becomes *exact*. Thus, focusing on the thermodynamic limit ($\mu \gg \hbar\omega$), we proceed as follows: semi-classically, the single-particle

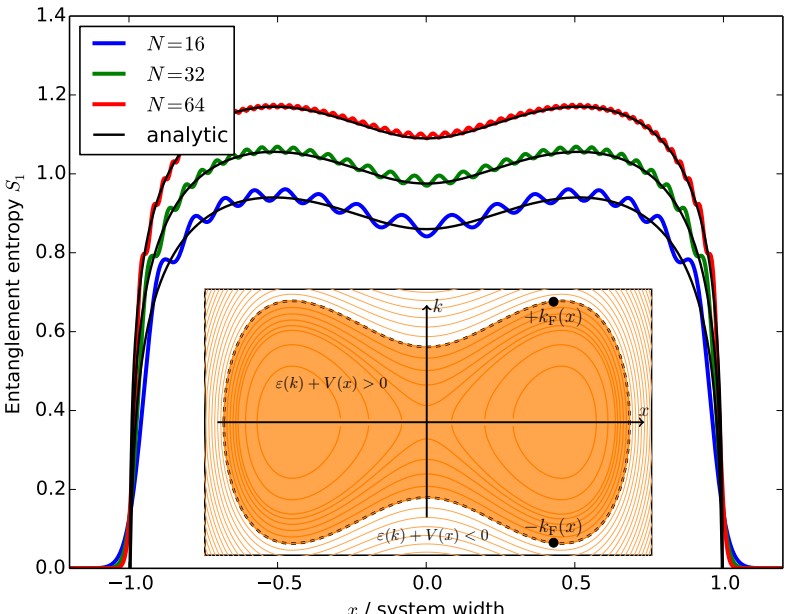

Figure 1: An example: a Fermi gas trapped in a double-well potential. To obtain non trivial results, we scaled the potential with the number $N$ of particles so as to have a finite density inside each well. Our analytic prediction for the entanglement entropy is checked against numerical data for $n = 1$: the agreement is excellent, and improves further when increasing $N$. Inset: in phase space, the quasi-classical orbitals are equipotentials enclosing an area that is an integer multiple of $2\pi$.

eigenstates correspond to the equipotentials $(x, k)$ that satisfy the Bohr-Sommerfeld quantization rule. To get the many-body ground state, one simply fills up all eigenstates with negative energies. The equipotentials can be locally parametrized as $(x, \pm k_{\mathrm{F}}(x))$, where

$$k_{\mathrm{F}}(x) = \sqrt{2(\mu - V(x))}. \tag{19}$$

The position-dependent Fermi velocity is $v_{\mathrm{F}}(x) = k_{\mathrm{F}}(x)$. The metric that underlies the problem is $ds^2 = dx^2 + v_{\mathrm{F}}(x)^2 dy^2$, one can obtain a complex coordinate system on the worldsheet just like before; the result for a general potential $V(x)$ reads

$$z(x, y) = \int^x \frac{dx'}{v_{\mathrm{F}}(x')} + iy, \qquad e^\sigma = v_F(x). \tag{20}$$

Notice that the real part of $z(x, y)$ is the time that a massless excitation takes to arrive at point $x$, starting from some reference point $x_0$ with $\mathrm{Re}\, z(x_0, y) = 0$. The coordinate $z$ always lives on an infinite strip $[\tau_1, \tau_2] \times \mathbb{R}$, whose width $\tau_2 - \tau_1$ depends on $V(x)$ and $\mu$, and which can be conformally mapped onto the upper half-plane by $z \mapsto g(z) = e^{i\pi \frac{z - \tau_1}{\tau_2 - \tau_1}}$. Correlation functions are then evaluated exactly as in the harmonic case above. Formula (16) holds, and can be used to derive new exact results for the entanglement entropy. For instance, considering $V(x) \propto |x|^p$ one recovers the results from the so-called trap size scaling [31].

One example that, to the best of our knowledge, cannot be solved with other means is that of a double well potential

$$V(x) = \alpha_4 x^4 - \alpha_2 x^2, \qquad \alpha_2, \alpha_4 > 0. \tag{21}$$

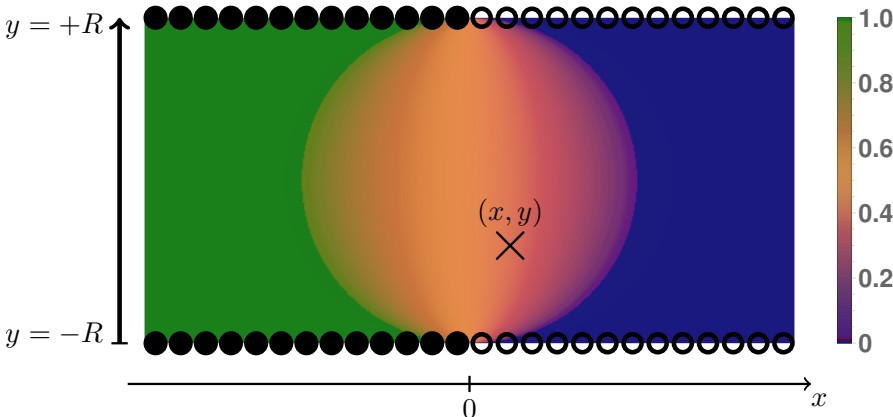

Figure 2: Imaginary time density profile corresponding to the quench from a semi-infinite box packed with fermions (filled black circles on the picture, the holes are shown with empty circles). The density at coordinate $(x, y)$ is defined as (25). Green colors correspond to a density close to one, blue to a density close to zero. In both cases the system is said to be "frozen": observables do not fluctuate at all. Intermediate colors correspond to finite densities, and have non trivial fluctuations. The fluctuating region is a disk of radius $R$, with density given by (25).

In Fig. 1 we compare some exact finite size calculations for this potential with our novel prediction. The figure shows that (apart from well-known finite size oscillations [32]) exact numerical results match perfectly our CFT prediction.

## 4 A non-equilibrium situation

We now show how the above framework can be adapted to deal with out-of-equilibrium situations. The most natural problem to attack is the Fermi gas (1) released from a harmonic trap. However, this problem may be solved by other methods, and it is known, for instance, that various observables obey a dynamical symmetry [33], including the entanglement entropy [30,34,35]. This symmetry relates time-dependent quantities to their time-independent, ground-state, counterpart, simply by rescaling the coordinate $x$ to $x/\sqrt{1 + \omega^2 t^2}$. Since the dynamical symmetry allows to deal with this problem in an efficient way, it is not the most illustrative example for our purposes.

Instead, we turn to a lattice gas, released from a semi-infinite box: the initial state is such that all sites $x < 0$ are filled, and all sites $x \geq 0$ are empty (this is also known as a quench from a domain wall initial state [36–41]). At $t > 0$, one lets the system evolve with the Hamiltonian

$$H = -\frac{1}{2} \sum_{x \in \mathbb{Z}} \left[ c_x^\dagger c_{x+1} + c_{x+1}^\dagger c_x \right].$$ (22)

In Fourier space the Hamiltonian is (up to an additive constant)

$$H = \int_{-\pi}^{\pi} \frac{dk}{2\pi} \varepsilon(k) c_k^\dagger c_k,$$ (23)

with dispersion relation $\varepsilon(k) = -\cos k$.

The relevant regime for an effective field theory description is that of large distances and late times, in a way such that the ratio $x/t$ is kept finite. In this limit the density profile at

time $t$ is [36]

$$\rho(x,t) = \frac{1}{\pi}\arccos\frac{x}{t}.$$ (24)

Again, the question we want to answer is: *what is the effective theory that captures long-range correlators* $\langle \phi_1(x_1,t_1)\dots\phi_n(x_n,t_n)\rangle$ *in this inhomogeneous system?* We expect that it should be a (lorentzian) Dirac theory in curved 1+1D spacetime. There are technical issues, however, associated with the lorentzian formulation of the problem—for instance, the metric would be degenerate, $ds^2 = (dx - \frac{x}{t}dt)^2$, and there would be no clear distinction between right- and left-movers—, so we chose to look at the problem in imaginary time, as routinely done in quench problems in CFT [18]. In this imaginary time approach to quantum quenches, the initial state becomes a boundary condition on the two sides of an infinite strip of width $2R$ in imaginary time direction $y$ [18,42,43]. Real-time correlators are recovered by first performing a Wick rotation $y \to it$, and then sending $R \to 0$. We focus on correlators $\langle\phi_1(x_1,y_1)\dots\phi_n(x_n,y_n)\rangle_R$, where $y_j \in [-R,R]$ (see Fig. 2 for the application to the domain wall quench). For example, the imaginary-time density profile is [44]

$$\langle\rho(x,y)\rangle_R \ = \ \frac{\langle\psi|e^{-(R-y)H}c_x^\dagger c_x e^{-(R+y)H}|\psi\rangle}{\langle\psi|e^{-2RH}|\psi\rangle} = \frac{1}{\pi}\arccos\frac{x}{\sqrt{R^2-y^2}},$$ (25)

which gives back the real-time profile (24) after performing the Wick rotation $y \to it$ and taking the limit $R \to 0$. The density is different from zero or one only inside the disc $x^2 + y^2 < R^2$; thus, there is a phase separation phenomenon, known as *arctic circle* [45]. This is shown in Fig. 2. In Ref. [44], the field theory that describes long-range correlations inside the disc was unraveled: it is a Dirac theory in curved space, with euclidean metric

$$ds^2 = dx^2 + 2\frac{xy}{R^2-y^2}dxdy + \frac{R^2-x^2}{R^2-y^2}dy^2.$$ (26)

Once we know this, it is again a straightforward exercise in CFT to compute correlation functions. As an illustration, we calculate again the entanglement entropy, for the bipartition $A\cup B$ with $A = [-\infty, x]$, $B = [x, +\infty]$ at times $t > x$ (without loss of generality we assume $x > 0$). This can be done by performing elementary manipulations similar to those of Sec. 3.1 The main difference with the previous calculation is that, due to the presence of a finite lattice spacing, Eq. (16) needs to be modified. Namely, the position-dependent cut-off is no longer simply proportional to $k_F$. In the *homogeneous* problem, it is known from the exact lattice solution [46] that the cut-off enters the formula for the Renyi entropies as $\sin(k_F)$, instead of $k_F$ in the continuous gas. As a consequence of separation of scales, in the inhomogeneous case, we thus need to replace the local cut-off $k_F(x,y)$ in Eq. (16) by $\sin(k_F(x,y))$. This leads to the following formula for the Renyi entropies in imaginary time,

$$S_n = \frac{n+1}{12n}\ln\left[\sin(k_F(x,y))e^{\sigma(z,\bar{z})}\left|\frac{dg(z)}{dz}\right|^{-1}\mathrm{Im}\,g(z)\right].$$ (27)

Here the complex coordinate $z = z(x,y)$ must be compatible with the conformal structure of the metric (26), *i.e.* one must have $ds^2 = e^{2\sigma}dzd\bar{z}$, for some function $e^\sigma$. Both $z$ and $e^\sigma$ are given in Ref. [44],

$$z(x,y) = \arccos\frac{x}{\sqrt{R^2-y^2}} - i\,\mathrm{arcth}\frac{y}{R}, \qquad e^\sigma = \sqrt{R^2-x^2-y^2}.$$ (28)

The function $(x,y) \mapsto z$ maps the interior of the disc $x^2 + y^2 < R^2$, namely the interior of the critical region in the $(x,y)$ plane, onto the infinite strip $[0,\pi]\times\mathbb{R}$. The strip itself can be sent

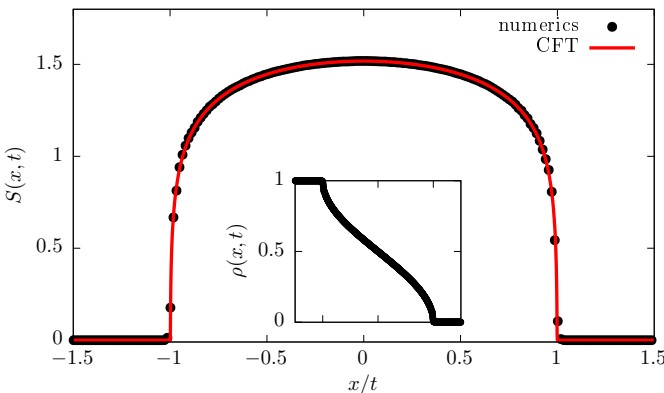

Figure 3: von Neumann entanglement entropy as a function of $x/t$ after a quench from the domain wall initial state. The numerical simulations are performed using a finite system with 4096 sites. Data for $t = 512$ are reported (black circles) and compared to our analytical prediction (red line): the agreement is nearly perfect. Inset: corresponding density profile $\rho(x, t)$ after the quench.

to the upper half plane by the conformal map $g(z) = e^{iz}$. Finally, recalling that [44]

$$k_F(x, y) = \text{Re}[z(x, y)] = \arccos \frac{x}{\sqrt{R^2 - y^2}}, \tag{29}$$

elementary algebraic manipulations and the use of (27) lead to the following expression for the Renyi entropies in imaginary time

$$S_n(x, y) = \frac{n+1}{12n} \ln\left[\frac{(R^2 - x^2 - y^2)^{3/2}}{R^2 - y^2}\right]. \tag{30}$$

The analytic continuation to real time is obtained by first substituting $y \to it$, and then sending $R \to 0$. This gives

$$S_n(x, t) = \frac{n+1}{12n} \ln\left[t(1 - (x/t)^2)^{3/2}\right], \tag{31}$$

a formula which was guessed from numerics in Ref. [48], and was calling for an analytic derivation. We just provided this derivation, which crucially relies on the metric (26) that underlies the whole problem. In Figure 3, we report a comparison of this prediction with numerical data and the agreement is perfect, up to the usual finite-size effects. The entanglement entropy for other bipartitions can be calculated as well, but the resulting formulas are more complicated and are therefore deferred to the appendix B.2. We mention that it is also possible to study different dispersion relations, but the results are rather technical; they are reported in appendix C.

# 5 Conclusion

Inhomogeneous 1D quantum systems are difficult to tackle and this is motivating an enormous activity in order to provide exact results in some regimes, as for example the recently developed integrable hydrodynamics [49,50] (anticipated in [47]) which may have important ramifications into transport in 1D systems [51,52] such as quantum wires. To shed some light on these timely problems, we have shown here, with a few free fermion examples, that CFT

methods may be extended to attack this class of problems. The key assumption of our work is separation of scales: the system is locally homogeneous (but only locally), which is also the working limit of all approaches constructed so far [47, 49, 50]. In this regime, one can write a field theory action with varying parameters. In the examples tackled in this paper, we found that the metric tensor should vary, leading to CFT in curved space. In particular, this new approach allows us to compute in a simple manner the entanglement entropies of these inhomogeneous systems (both in- and out-of-equilibrium) which are otherwise very difficult (in most cases impossible) to obtain with other methods. It is important to stress that the background metric and the inhomogeneous cut-off are non-universal functions and must be viewed as *inputs* of the formalism. They should be obtained a priori by a proper microscopic computation.

We believe anyway that the results of this paper should open the door to several new developments. For example, by following the CFT approach of Ref. [53], our method could be used not only to determine the entanglement entropy, but also the entanglement negativity which is a proper measure of entanglement in mixed states [54] and it is not yet known how to compute it exactly for free fermionic systems. Another interesting development would be to recover by random matrix techniques [28, 55] the results we obtained for the entanglement entropy in a harmonic potential, possibly unveiling new structures of the random matrices. Having exact results also for arbitrary trapping potential could also help understanding whether these general cases could be tackled by random matrix techniques.

From a physical point of view, the main open problem is how to describe inhomogeneous interacting 1D systems, most importantly Luttinger liquids such as Heisenberg spin chains and Lieb-Liniger gases. In these cases, one expects also the coupling constant (or Luttinger parameter $K$) to vary with position in spacetime. Fixing such parameters is a challenging problem. We hope that this paper will stimulate activity in that direction.

## Acknowledgements

We thank Nicolas Allegra and Masud Haque for collaboration on closely related topics [41,44], and Viktor Eisler for useful discussions.

**Funding information**   This work was supported partially by the ERC under Starting Grant 279391 EDEQS (PC), and by the CNRS Interdisciplinary Mission and Région Lorraine (JD). JD, JMS, and JV thank SISSA for hospitality.

## A   The Dirac action in curved space-time

Here we explain the form of the Dirac action in a curved space-time, that was used in the main text. The most generic form of the Dirac action in a curved two-dimensional eulidean space-time is given by

$$S = \frac{1}{2\pi} \int d^2 x \sqrt{g} \, e_a^\mu \left[ \bar{\Psi} \gamma^a \left( \frac{1}{2} \overleftrightarrow{\partial}_\mu + i A_\mu^{(v)} + i A_\mu^{(a)} \gamma_5 \right) \Psi \right], \tag{32}$$

where we chose $\gamma_1 = -\sigma_2$, $\gamma_2 = \sigma_1$, $\gamma_5 = \sigma_3$ and $\bar{\Psi} = \Psi^\dagger \gamma_2$. The $\sigma_\alpha$ are the usual Pauli matrices. In this representation the two components of the spinor $\Psi$ are the chiral components $\psi_R$ and $\psi_L$; the function $A_\mu^{(v)}$ and $A_\mu^{(a)}$ are the vector and axial gauge field associated with the $U(1)$ gauge symmetry and the (anomalous) chiral symmetry. The matrix $e_a^\mu$ is called a tetrad: it

maps the tangent space of the manifold into $\mathbb{R}^2$ while preserving the inner product. It satisfies $g_{\mu\nu}e_a^\mu e_b^\nu = \delta_{ab}$, where $g_{\mu\nu}$ is the metric tensor and $\delta_{ab}$ the flat metric. In two dimensions, it is convenient to use complex coordinates $z = x^1 + ix^2$ and $\bar{z} = x^1 - ix^2$. The metric is conformally flat and off-diagonal, the line element thus reads $ds^2 = e^{2\sigma}dz\,d\bar{z}$. The tetrad is diagonal if one chooses complex coordinates both in the tangent space and in $\mathbb{R}^2$. Its only non-vanishing components are complex conjugated and have modulus $e^{-\sigma}$. Using finally $\sqrt{g} = e^{2\sigma}$ we can then rewrite the Dirac action in complex coordinates as

$$S = \frac{1}{2\pi}\int d^2z\, e^{\sigma+i\theta}[\psi_R^\dagger(\overleftrightarrow{\partial_{\bar{z}}} + iA^{(v)} + iA^{(a)})\psi_R] + \frac{1}{2\pi}\int d^2z\, e^{\sigma-i\theta}[\psi_L^\dagger(\overleftrightarrow{\partial_z} + i\bar{A}^{(v)} - i\bar{A}^{(a)})\psi_L], \tag{33}$$

where $A^{(v/a)}$ and $\bar{A}^{(v/a)}$ are the complex $z,\bar{z}$ components of the vector and axial potential. The rotation of angle $\theta$ in (33) between the tangent space of the manifold and the flat euclidean space does not alter any of the conformal maps discussed in the main text. This holds because the twist fields needed to compute the entropy are spinless [27]. The extra phase factors in the fermionic propagators may be restored by performing a $U(1)$ gauge transformation and a chiral transformation. The former acts on the chiral fermions as $\psi_{R/L} \to e^{i\varphi}\psi_{R/L}$, and on the gauge field as $A^{(v)} \to A^{(v)} - \partial_{\bar{z}}\varphi$, $\bar{A}^{(v)} \to \bar{A}^{(v)} - \partial_z\varphi$; the chiral transformation instead acts as $\psi_{R/L} \to e^{\pm i\varphi}\psi_{R/L}$, and the gauge fields get modified as $A^{(a)} \to A^{(a)} - \partial_{\bar{z}}\varphi$ and $\bar{A}^{(a)} \to \bar{A}^{(a)} - \partial_z\varphi$.

## B The entanglement entropies of a finite interval $[x_1, x_2]$

### B.1 The harmonically trapped Fermi gas

The entropies of a finite interval $[x_1, x_2]$ are slightly more complicated to calculate than the case of the semi-infinite line (cf. Eq. (17)). Following Ref. [21], the Renyi entanglement entropy of order $n$ is related to the two-point function of the twist field $\mathcal{T}_n$ on the desired geometry. Here we work out the generalization to inhomogeneous systems and in particular for the case of fermions in a harmonic trap. In this case, the two-point function can be related to the one of the upper half plane by the combined action of a Weil transformation and the conformal mapping $g(z) = e^{i(z+\pi/2)}$ from the strip $[-\frac{\pi}{2}, \frac{\pi}{2}] \times \mathbb{R}$ to the upper half-plane. This allows us to write such two-point function in the $z$ coordinate as

$$\langle \mathcal{T}_n(z_1)\tilde{\mathcal{T}}_n(z_2) \rangle = \left(e^{-\sigma(z_1)}\left|\frac{dg(z_1)}{dz_1}\right|\right)^{\Delta_n}\left(e^{-\sigma(z_2)}\left|\frac{dg(z_2)}{dz_2}\right|\right)^{\Delta_n}\langle \mathcal{T}_n(g(z_1))\tilde{\mathcal{T}}_n(g(z_2)) \rangle_{\text{uhp}}. \tag{34}$$

The field $\tilde{\mathcal{T}}_n$ is the conjugated twist field [21, 27]. The calculation of two-point functions of twist fields in the upper half plane is, in general, a very challenging problem (indeed by images trick, it can be turned into a four point function in the complex plane which have been considered e.g. in Refs. [56, 57]). However, in the case of the massless free fermion field-theory, this two-point function in the half plane simplifies considerably and our desired object boils down to [30, 58] (up to unimportant multiplicative constants)

$$\langle \mathcal{T}_n(z_1)\tilde{\mathcal{T}}_n(z_2) \rangle = \left[e^{-\sigma(z_1)}\left|\frac{dg(z_1)}{dz_1}\right|e^{-\sigma(z_2)}\left|\frac{dg(z_2)}{dz_2}\right|\frac{|g^*(z_1) - g(z_2)|^2}{\text{Im}(g(z_1))\text{Im}(g(z_2))|g(z_1) - g(z_2)|^2}\right]^{\Delta_n}. \tag{35}$$

In our setup $z_1 = \arcsin(x_1/L)$ and $z_2 = \arcsin(x_1/L)$. Using again $g(z) = e^{i(z+\pi/2)}$ and (11), we obtain

$$
\begin{aligned}
S_n(x_1, x_2) &= S_n(x_1) + S_n(x_2) + \frac{n+1}{12n} \ln \left| \frac{\sqrt{1-\frac{x_1^2}{L^2}} - \sqrt{1-\frac{x_2^2}{L^2}} + i\frac{(x_1-x_2)}{L}}{\sqrt{1-\frac{x_1^2}{L^2}} + \sqrt{1-\frac{x_2^2}{L^2}} + i\frac{(x_1-x_2)}{L}} \right|^2 \\
&= \frac{n+1}{6n} \ln \left[ \frac{2L^3 \left(1-(\frac{x_1}{L})^2\right)^{3/4} \left(1-(\frac{x_2}{L})^2\right)^{3/4} \left(1 - \frac{x_1 x_2}{L^2} - \sqrt{\left(1-\frac{x_1^2}{L^2}\right)\left(1-\frac{x_1^2}{L^2}\right)}\right)}{(x_1 - x_2)} \right],
\end{aligned}
$$

(36)

where $S_n(x_i)$ is given in Eq. (17). This is the result reported in the main text.

## B.2 The domain-wall quench

For the case of a finite interval, the Renyi entanglement entropy are easily obtained by using Eq. (35) and the conformal mapping from the strip $[0, \pi] \times \mathbb{R}$ to the upper half-plane $g(z) = e^{iz}$. Setting $y_1 = y_2 \equiv y$ (i.e. equal imaginary times) and using (28), we get

$$
\begin{aligned}
S_n(x_1, x_2, y) &= S_n(x_1, y) + S_n(x_2, y) + \frac{n+1}{12n} \ln \left| \frac{\sqrt{R^2 - x_1^2 - y^2} - \sqrt{R^2 - x_2^2 - y^2} - i(x_1 - x_2)}{\sqrt{R^2 - x_1^2 - y^2} + \sqrt{R^2 - x_2^2 - y^2} - i(x_1 - x_2)} \right|^2 \\
&= \frac{n+1}{6n} \left[ \ln \frac{(R^2 - x_1^2 - y^2)^{3/4} (R^2 - x_2^2 - y^2)^{3/4}}{(R^2 - y^2)^{3/2}} + \right. \\
&\qquad \left. + \ln \frac{\left(R^2 - x_1 x_2 - y^2 - \sqrt{(R^2 - x_1^2 - y^2)(R^2 - x_2^2 - y^2)}\right)}{(x_1 - x_2)} \right].
\end{aligned}
$$

(37)

Analytically continuing the result (37) to real time and using the rescaled variables $\zeta_i = x_i/t$ we obtain

$$
S_n(x_1, x_2, t) = \frac{n+1}{6n} \ln \left[ \frac{t(1-\zeta_1^2)^{3/4}(1-\zeta_2^2)^{3/4}(1 - \zeta_1\zeta_2 - \sqrt{(1-\zeta_1^2)(1-\zeta_2^2)})}{\zeta_1 - \zeta_2} \right].
$$

(38)

This formula is valid in the regime $t > |x_1|, |x_2|$. Notice that this is very similar to Eq. (36), and indeed it has the same dependence of the rescaled variables $\zeta_i$. However, the leading dimensional term in one case scales like $\ln t$ while in the other as $\ln L^2$ (times $(n+1)/(6n)$ in both cases). This is somehow reminiscent of a similar anomalous scaling found in local quenches [59] and it is unclear whether there is a connection between the two.

# C  The domain wall quench for a more general dispersion relation

## C.1 Imaginary time treatment

Here we come back to the entropy of a single interval $(-\infty; x]$ following a domain wall quench, but with a more general dispersion relation. We consider Hamiltonian (23) with a dispersion relation of the form

$$
\varepsilon(k) = -\sum_{n \geq 1} a_{2n+1} \cos(2n+1)k,
$$

(39)

which contains only odd Fourier modes. The reason for choosing this special form is mainly technical. As we shall see it makes explicit computations much simpler due to the symmetries $\varepsilon(-k) = \varepsilon(k)$ and $\varepsilon(\pi - k) = -\varepsilon(k)$.

In real time the stationary phase equation governing the long time dynamics is [41]

$$v(k) = \frac{d\varepsilon(k)}{dk} = \frac{x}{t}, \tag{40}$$

and may be used to compute systematically all correlation functions. Now, due to the form (39), if $k_s(x,t)$ is a solution, then so is $\pi - k_s(x,t)$. In the following we will assume that there are at most two solutions $k_s(x,t)$ and $\pi - k_s(x,t)$ to the previous equation. Under these assumptions, the density profile is

$$\rho(x,t) = \frac{k_F(x,t)}{\pi} = \frac{\pi - 2k_s(x,t)}{2\pi} = 1/2 - \frac{k_s(x,t)}{\pi} \tag{41}$$

As is emphasized in the main text and above, it is convenient to think of this problem in imaginary time. The stationary phase equation for the imaginary time problem reads [44]

$$\frac{d\Phi_{x,y,R}}{dk} = 0, \qquad \Phi_{x,y,R}(k) = kx + iy\varepsilon(k) + R\tilde{\varepsilon}(k), \tag{42}$$

with

$$\tilde{\varepsilon}(k) = -\sum_{n \geq 1} a_{2n+1} \sin(2n+1)k, \tag{43}$$

the Hilbert transform of $\varepsilon(k)$. In case there is only a finite number of Fourier modes in the dispersion, solving (42) amounts to solving algebraic equations in $e^{ik}$, which can be done in principle systematically. Now we make the extra assumption that there are only two solutions $k = z(x,y)$ and $k = -z^*(x,y)$. The new variable $z$ lives in an infinite strip $0 \leq \operatorname{Re} z \leq \pi$. The metric then reads

$$ds^2 = e^{\sigma(x,y)} dz d\bar{z}, \qquad e^{\sigma(x,y)} = \left. \frac{d^2\Phi_{x,y,R}(k)}{dk^2} \right|_{k=z(x,y)}. \tag{44}$$

The conformal distance to the boundary may be obtained by mapping the infinite strip to the upper half plane. Such a mapping is once again provided by $g(z) = e^{iz}$, so we obtain

$$d(x,y) = e^{\sigma(x,y)} \left| \frac{dg}{dz} \right|^{-1} \operatorname{Im} g(z) = e^{\sigma(x,y)} \sin(\operatorname{Re} z(x,y)). \tag{45}$$

Separating the real and imaginary parts $z(x,y) = \kappa(x,y) + iQ(x,y) = \kappa + iQ$, the above reads

$$d(x,y) = \left( \left. \frac{d^2\Phi_{x,y,R}(k)}{dk^2} \right|_{k=\kappa+iQ} \right) \sin \kappa, \tag{46}$$

where $\kappa$ and $Q$ satisfy the system of equations

$$\sum_n (2n+1) a_{2n+1} \cos(2n+1)\kappa \left[ y \sinh(2n+1)Q + R \cosh(2n+1)Q \right] = x, \tag{47}$$

$$\sum_n (2n+1) a_{2n+1} \sin(2n+1)\kappa \left[ y \cosh(2n+1)Q + R \sinh(2n+1)Q \right] = 0. \tag{48}$$

The reasoning to get the entanglement entropy should be clear at this point: first we get $\kappa$ and $Q$ from the above set of equations, and then we use them to compute the conformal distance

to be plugged in the appropriate equation for the entropy. For example in the case of the standard cosine dispersion relation, using this procedure, we find the already known result $d(x, y) = (R^2 - x^2 - y^2)/\sqrt{R^2 - y^2}$, which can then be continued to real time $y = it, R \to 0$. It is important to stress that the concept of conformal distance only makes sense in imaginary time: in principle only after computing $d(x, y)$ in imaginary time we are allowed to make the analytic continuation to real time.

In practice, one can avoid finding the general solution of the above system of equations. Indeed, because of their analytic structure, we can perform the analytic continuation at the level of Eqs. (47) and (48), relaxing the requirement that $\kappa$ and $Q$ are real. Plugging $y = it$ and taking the limit $R \to 0$, we find that $Q = i\pi/2$ is a trivial solution to (48), due to the absence of even Fourier modes. The variable $\kappa$ is then the solution of

$$\frac{x}{t} = \sum_{n \geq 1} (-1)^{n+1} \cos[(2n+1)\kappa] = v(\kappa + \pi/2), \tag{49}$$

and so we get $\kappa = k_s - \pi/2$. The analytic continuation of the conformal distance becomes

$$\lim_{R \to 0} d(x, it) = t \left( \left. \frac{dv}{dk} \right|_{k=k_s} \right) \cos k_s = t v'(k_s) \cos k_s. \tag{50}$$

Putting back the contribution from the UV cutoff, $\sin k_F = \cos k_s$, we finally obtain

$$S_n(x, t) = \frac{1}{12} \left( 1 + \frac{1}{n} \right) \ln \left( t \, v'(k_s) \cos^2 k_s \right). \tag{51}$$

Let us briefly comment on the case with more than two solutions to the stationary phase equation (40). With the form (39) of the dispersion they always come in pairs. We label them as $k_s^{(i)}$ and $\pi - k_s^{(i)}$ for $i = 1, \ldots, p$. $p$ may be interpreted as a number of Fermi seas. As before we compute the entanglement entropy using field theory, and add to this the contribution coming from the position-dependent UV cutoff. The generalization of our field-theoretical framework to account for several Fermi seas is straightforward: we simply introduce $p$ different species of Dirac Fermions, one for each Fermi sea (pair of stationary modes). Being non-interacting particles, all the Dirac species contribute independently to the entanglement entropy. Obtaining the position-dependent cutoff is more tricky, but can nevertheless be done using exact equilibrium results obtained for several Fermi seas[1]. We refer to Ref. [60] for the details. The lattice cutoff takes now the more complicated form

$$f(k_s^{(1)}, \ldots, k_s^{(p)}) = \prod_{i=1}^{p} \cos k_s^{(i)} \prod_{1 \leq i < j \leq p} \left| \frac{\sin \frac{k_s^{(i)} - k_s^{(j)}}{2}}{\cos \frac{k_s^{(i)} + k_s^{(j)}}{2}} \right|^{(-2)^{i-j+1}}. \tag{52}$$

The final result for the entropy reads

$$S_n(x, t) = \frac{1}{12} \left( 1 + \frac{1}{n} \right) \left\{ \sum_{i=1}^{p} \ln \left( t \, v'(k_s^{(i)}) \cos^2 k_s^{(i)} \right) - 2 \sum_{1 \leq i < j \leq p} (-1)^{i-j} \log \left| \frac{\sin \frac{k_s^{(i)} - k_s^{(j)}}{2}}{\cos \frac{k_s^{(i)} + k_s^{(j)}}{2}} \right| \right\}. \tag{53}$$

## C.2  An explicit example

Let us now work out an example where the formula (51) may be computed explicitly. We consider a dispersion of the form

$$\varepsilon(k) = -\cos k + \frac{\alpha}{9} \cos(3k), \qquad 0 < \alpha < 1. \tag{54}$$

---

[1]We are grateful to Viktor Eisler for pointing that out to us.

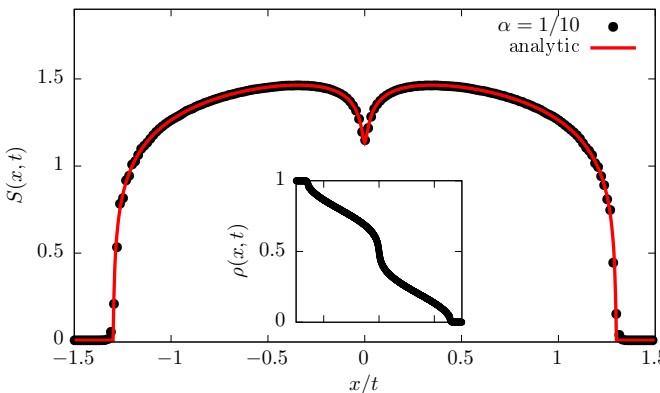

Figure 4:  von Neumann entanglement entropy as a function of $x/t$ after a quench from the domain wall initial state, for dispersion relation $\varepsilon(k) = -\cos k + \alpha \cos 3k$ and $\alpha = 1/10$. he numerical simulations are performed using a finite system with 4096 sites. Data for $t = 512$ are reported (black circles) and compared to our analytical prediction (red lines): the agreement is nearly perfect. Inset: corresponding density profile $\rho(x, t)$ after the quench.

The restriction on $\alpha$ ensures that there are at most two solutions to the stationary phase equation, $k_s(x, t)$ and $\pi - k_s(x, t)$. $k_s$ may be obtained by solving a cubic equation:

$$k_s(x, t) = \arcsin \left[ \frac{1 - \alpha - B(x, t)^{2/3}}{2\sqrt{\alpha}\,B(x, t)^{1/3}} \right], \qquad B(x, t) = \sqrt{9\alpha(x/t)^2 + (1 - \alpha)^3} - 3\sqrt{\alpha}(x/t).$$
(55)

The density profile in the inhomogeneous region $|x/t| \le 1 + \alpha/3$ is then given by

$$\rho(x, t) = \frac{1}{\pi} \arccos \left[ \frac{1 - \alpha - B(x, t)^{2/3}}{2\sqrt{\alpha}\,B(x, t)^{1/3}} \right],$$
(56)

and the entropy is obtained by plugging (55) in (51). In Fig. 4 we report the numerical data for the von Neumann entanglement entropy for the dispersion relation with $\alpha = 0.1$: this shows perfect agreement with our prediction.

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
