# Peer review of "Conformal Field Theory for Inhomogeneous One-dimensional Quantum Systems: the Example of Non-Interacting Fermi Gases"

_SciPost Physics, doi:SciPost Phys. 2, 002 (2017)_

## Round 2 · Referee Report · Anonymous · 2016-12-20

Strengths

1- Solidity of the arguments
2- Relevance of the topic
3- Check of the arguments against previous results and numerics

Weaknesses

1- Novelty: some of the ideas and computations appeared already in [41],[44]

Report

The authors discuss the applicability of conformal field theory to
inhomogeneous free fermionic systems in one dimension. Based on a
plausible scaling argument and previous results by some of the
authors, they derive an effective description of the problem in terms
of a Dirac fermion on a curved space-time. This description is then
tested against previous random matrix theory results and
numerics. Compelling evidence for its applicability to the computation
of entanglement entropies is obtained. I find the presentation
accessible and the topic of interest for the readership of SciPost
physics. Therefore, I recommend this paper for publication, upon the
discussion of the minor changes and comments listed below.

Requested changes

1- page 2, line 4:
"such diverse cases as" -> "diverse cases such as"

2- page 4, below Eq 4: the transformation law of psi_L should be added

3- page 5, line 1: v(k_F(x)) -> v_F(k_F(x))

4- The authors could write the boundary conditions for the theory of Eq. 7.

5- page 6, the sentence
"The conformal mapping ... upper half plane."
is redundant and can be removed.

6- page 7, fig 1 is supposed to describe the potential of Eq. 21.
What are the values of alpha2 and alpha4?

7- In section 3, the authors generalize to the case of an arbitrary
potential V. It is not clear to me why z should always live on the
strip tau1, tau2. Should not this depend on the support of the
density of states? Thinking about random matrix theory suggests a
scenario with several cuts in general.

8- The method used requires large N, and it seems to me that the
authors do not address the corrections to their results, apart from
the sentence in parenthesis at the end of section 3. It might be
helpful to discuss a bit more this aspect, eventually reminding the
reader about the literature.

9- Fig 3: the legend says alpha=0. I think that the authors have in
mind the parameter of App C, Eq. 54, and this part of the legend
has to be removed.

10- 6th line after Eq. 32, e^\nu_a -> e^\nu_b .

11- 2nd line before Eq. 33, "with modulus" -> "modulus"

12- the right bracket ] in Eq. 33 should go after psi

13- Eq. 39, cos(2n+1)k -> cos[(2n+1)k]. Same in 43.

14- Below eq. 43, "In case there are only" -> "In case there is only"

---

## Round 2 · Referee Report · Anonymous · 2016-12-20

Strengths

1- Solidity of the arguments
2- Relevance of the topic
3- Check of the arguments against previous results and numerics

Weaknesses

1- Novelty: some of the ideas and computations appeared already in [41],[44]

Report

The authors discuss the applicability of conformal field theory to
inhomogeneous free fermionic systems in one dimension. Based on a
plausible scaling argument and previous results by some of the
authors, they derive an effective description of the problem in terms
of a Dirac fermion on a curved space-time. This description is then
tested against previous random matrix theory results and
numerics. Compelling evidence for its applicability to the computation
of entanglement entropies is obtained. I find the presentation
accessible and the topic of interest for the readership of SciPost
physics. Therefore, I recommend this paper for publication, upon the
discussion of the minor changes and comments listed below.

Requested changes

1- page 2, line 4:
"such diverse cases as" -> "diverse cases such as"

2- page 4, below Eq 4: the transformation law of psi_L should be added

3- page 5, line 1: v(k_F(x)) -> v_F(k_F(x))

4- The authors could write the boundary conditions for the theory of Eq. 7.

5- page 6, the sentence
"The conformal mapping ... upper half plane."
is redundant and can be removed.

6- page 7, fig 1 is supposed to describe the potential of Eq. 21.
What are the values of alpha2 and alpha4?

7- In section 3, the authors generalize to the case of an arbitrary
potential V. It is not clear to me why z should always live on the
strip tau1, tau2. Should not this depend on the support of the
density of states? Thinking about random matrix theory suggests a
scenario with several cuts in general.

8- The method used requires large N, and it seems to me that the
authors do not address the corrections to their results, apart from
the sentence in parenthesis at the end of section 3. It might be
helpful to discuss a bit more this aspect, eventually reminding the
reader about the literature.

9- Fig 3: the legend says alpha=0. I think that the authors have in
mind the parameter of App C, Eq. 54, and this part of the legend
has to be removed.

10- 6th line after Eq. 32, e^\nu_a -> e^\nu_b .

11- 2nd line before Eq. 33, "with modulus" -> "modulus"

12- the right bracket ] in Eq. 33 should go after psi

13- Eq. 39, cos(2n+1)k -> cos[(2n+1)k]. Same in 43.

14- Below eq. 43, "In case there are only" -> "In case there is only"

  • validity: high
  • significance: high
  • originality: good
  • clarity: good
  • formatting: excellent
  • grammar: excellent

Author:  Jean-Marie Stéphan  on 2017-02-13  [id 94]

(in reply to Report 1 on 2016-12-20)
Category:
answer to question

We thank the referee for his/her critical reading of the manuscript. We would like to object to the «weakness» mentioned by the referee. First, there is no overlap with the results of Ref. [41] at all; the referee is wrong about that. Second, there is indeed some overlap with Ref. [44], in the sense that we are using results from Ref. [44] to derive new results in the present paper. This is clearly explained in the text. We do not quite agree that using previous results of ours to derive new ones should be viewed as a weakness though; instead, this is just the usual way one makes progress in science. In fact, this paper should be viewed as one in a series of works that aim at unraveling the field theory description of
inhomogeneous 1+1d quantum systems; it’s very likely that there will be other subsequent papers
by us and our collaborators on similar topics.

We also thank the referee for catching up the typos; we have made corrections to the final version accordingly. We would like to answer points 7 and 8 raised by the referee.

"7- In section 3, the authors generalize to the case of an arbitrary
potential V. It is not clear to me why z should always live on the
strip tau1, tau2. Should not this depend on the support of the
density of states? Thinking about random matrix theory suggests a
scenario with several cuts in general."

In our setup, the region where the density is strictly positive coincides with the region
where $V(x)-\mu > 0$. We are just assuming that this is a single interval. We do not think that the case of multiple intervals would be much more interesting: it would just correspond to completely independent single-interval systems. This is not like the multi-cut case in random matrix theory, where there is some interesting interplay between the cuts (the connection between our problem and random matrix theory holds only for the harmonic trap).

"8- The method used requires large N, and it seems to me that the
authors do not address the corrections to their results, apart from
the sentence in parenthesis at the end of section 3. It might be
helpful to discuss a bit more this aspect, eventually reminding the
reader about the literature."

Indeed, we do not address the corrections in this paper. They can be tackled by combining standard ideas from homogeneous Luttinger liquid theory
(see for instance Ref. [1]) with the curved background metric from this paper. Such finite-N corrections are very interesting, however their study is more technical than what is done in this paper, and it probably deserves a paper on its own. We hope to come back to these corrections in future work.

---

## Round 2 · Referee Report · Anonymous · 2017-1-16

Strengths

-

Weaknesses

-

Report

The paper deals with the very interesting case of inhomogeneous quantum systems,
a case of general interest both for theoretical and experimental reasons. Even though
the authors focus their attention on rather simple systems (non-interacting Fermi gas
in external potentials), the final result is instead quite amusing and even rich (I am referring in particular
to the interesting comparison with random matrix theory and the total absence of any other
means to compute entanglement entropy). It should be seen whether the separation of energy
scales also persists in presence of interaction among the particles and in which way the interaction
modifies the picture put forward in this paper, if smoothly or radically. However the progress
made in this paper amply justifies its publication on SciPost.

Requested changes

-

---

## Round 2 · Referee Report · Anonymous · 2017-1-17

Strengths

1- very interesting ideas
2- clarity and conciseness of text
3- compelling numerical evidence for derived analytic expressions
4- new interesting formulae

Weaknesses

1- derivation lacking some discussion elements
2- currently method restricted to free fermion

Report

In this paper, the authors introduce a new method / new set of ideas in order to treat quantum many-body problems at criticality in weakly inhomogeneous potentials. The main idea is that, if the inhomogeneity length scale is much larger than the microscopic length scale, then at intermediate lengths, the system looks locally, at every point, like a homogeneous field theory (this is very similar to hydrodynamic ideas). It is then argued that in order to take into account the inhomogeneities, at least in the case of the free fermion, one simply needs to introduce a nontrivial curvature. This curvature is chosen in order to correctly reproduce locally the two-point function (this is sufficient in order to reproduce locally all correlation functions in free models), and then any quantity of interest can be evaluated by using this field theory. The example of the entanglement entropy is discussed, both at and out of equilibrium. The analytic results are strongly supported by numerics, with impressive agreement.

The paper is well written and original, and the results and ideas are interesting and extremely well supported by numerics. It is hoped that similar ideas can be used beyond the free fermion case, although there exists many potential obstacles, some pointed out by the authors. In any case, I think this paper is very good, and accept for publication in SciPost, after the comments below have been addressed (but I do not need to see the paper again, I have confidence that the authors will appropriately add necessary discussion elements).

A.

My main unease is with respect to the derivation of the fact that the Dirac theory in curved space is the correct large-scale theory. The results are most likely correct as numerics show, and the proposal is very natural. But I feel the derivation would benefit from some additional discussion points (if these are indeed better understood by the authors), especially in view of going beyond the free fermion.

It is well explained how the ground state of the Dirac theory in curved space reproduces the correct *local* two-point function of the microscopic models at intermediate length scales. However, some questions (all related) arise.

(1) The curved space is a natural way of introducing a scale and inhomogeneities in an otherwise conformal model. It is also naturally related to the space-dependent velocity that is observed locally. However, there are more ways; for instance, why not also introduce a space-dependent mass term? This would still reproduce the correct local (conformal-looking) two-point functions, as the mass would only act nontrivially at large distances. In free Dirac fermion models, this I believe at least in some cases this might be made equivalent to a curvature in a conformal model (I am not completely sure), but then how general is the assumption that a curvature is the correct way (i.e. what about other models)? Another way is to have a state, say in a conformal model, that has itself an intrinsic scale (a thermal state in usual CFT is such an example - what about a space-dependent temperature?). I think the "objection" here is that the derivation shows equality of certain quantities, but does not show explicitly that the curved-space theory emerges from a theory with a potential.

(2) This is related to (1). If we are just looking at local correlations, then the derivation in the paper shows that the curved-space theory is correct. Any other implementation of the inhomogeneities reproducing the same local fermion two-point function would lead to the same correct local theory, of course. But then the claim of the authors is that the curved-space theory is the correct *large-scale* theory: it reproduces nonlocal quantities such as entanglement (or two-point functions), not just locally but also at the length scale of the inhomogeneities. Why is that so? How do we go from correct local theory to correct large-scale theory? Can we say something about large-scale correlations (e.g. how they decay) a priori (i.e. from knowledge that microscopic model is at zero temperature), even in the presence of an inhomogeneous potential, that would forbid e.g. a mass term? This is not clear as the potential might forbid us from properly taking large distance limit in correlation functions (e.g. finite or harmonic well).

Why is a potential in the otherwise critical (low-energy) microscopic model only related to a change of geometry of the emerging field theory? The derivation is not constructive enough to allow us to understand this point.

B.

I also think it might be good to discuss cases near criticality. What would happen in models where the correlation length is not infinite, and comparable or smaller than the inhomogeneity length scale? Do we get a massive (or otherwise nonconformal) model on a curved space?

small thing: please define precisely $\rho$ in eq (5)

Requested changes

1- if possible add discussion elements as per report

  • validity: high
  • significance: high
  • originality: high
  • clarity: high
  • formatting: perfect
  • grammar: perfect

Author:  Jean-Marie Stéphan  on 2017-02-13  [id 95]

(in reply to Report 4 on 2017-01-17)

We thank the referee for his/her comments. Here we answer some of his/her concerns:

A.

« But I feel the derivation would benefit from some additional discussion points (if these are indeed better understood by the authors), especially in view of going beyond the free fermion. »

The reason we expect a Dirac theory is because the underlying microscopic model is a free fermion model with conserved number of particles. For a microscopic model of interacting particles, the theory would not be a Dirac theory. However, upon bosonization, it would be a free boson field with a renormalized coupling constant (or Luttinger parameter K), still in a curved metric. We are currently investigating this.

A.1) « why not also introduce a space-dependent mass term? »

Simply because we are describing a microscopic model that has algebraically decaying correlations, so we know in advance that there cannot be a mass term.

«This would still reproduce the correct local (conformal-looking) two-point functions, as the mass would only act nontrivially at large distances. »

This comment sounds wrong. A non-zero mass term would immediately imply exponentially decaying correlations, even at distances $|x-x’| \sim \ell$, where $\ell$ is the intermediate scale at which the system is a homogeneous 1d Fermi liquid. This is ruled out by direct calculation in the microscopic model, which shows that correlations decay algebraically.

« In free Dirac fermion models, this I believe at least in some cases this might be made equivalent to a curvature in a conformal model (I am not completely sure) »

The referee is right about the fact that, usually, finite curvature is associated to a finite length scale, and that correlations functions decay quickly (e.g. exponentially) at distances larger than this length scale. In that sense, the effect of finite curvature is similar to a finite mass. However, in the cases treated in this paper, we find that the curvature scales as $1/L^2$ where $L$ is the size of the system. The effect of curvature is then comparable to a finite-size correction to scaling, rather than to a finite mass.

« Another way is to have a state, say in a conformal model, that has itself an intrinsic scale (a thermal state in usual CFT is such an example - what about a space-dependent temperature?). »

Finite temperature can be ruled out for the same reason as a finite mass: it is not compatible with algebraically decaying correlation functions.

« I think the "objection" here is that the derivation shows equality of certain quantities, but does not show explicitly that the curved-space theory emerges from a theory with a potential. »

It is true that we checked our claim only for «certain quantities », namely the propagator itself. But since we are dealing with a free fermion model, the
propagator is sufficient to ensure that all higher-point correlation functions also match. In fact, it is possible to calculate the asymptotics of the propagator directly from the microscopic model. For instance, for the harmonic trap, the exact calculation boils down to using the asymptotics of Hermite polynomials at arbitrary distances (not only short distances). The result matches the prediction from field theory in curved metric.

A.2) « If we are just looking at local correlations, then the derivation in the paper shows that the curved-space theory is correct. Any other implementation of the inhomogeneities reproducing the same local fermion two-point function would lead to the same correct local theory, of course. But then the claim of the authors is that the curved-space theory is the correct *large-scale* theory: it reproduces nonlocal quantities such as entanglement (or two-point functions), not just locally but also at the length scale of the inhomogeneities. Why is that so? How do we go from correct local theory to correct large-scale theory? Can we say something about large-scale correlations (e.g. how they decay) a priori (i.e. from knowledge that microscopic model is at zero temperature), even in the presence of an inhomogeneous potential, that would forbid e.g. a mass term? This is not clear as the potential might forbid us from properly taking large distance limit in correlation functions (e.g. finite or harmonic well). »

Yes, it’s possible to go from short-range correlations to large-scale correlations. This is the most important point of the paper, actually. Again,a direct calculation at large distances is possible in the case of a harmonic trap, and the result matches perfectly the prediction from the massless Dirac theory in curved metric.

« Why is a potential in the otherwise critical (low-energy) microscopic model only related to a change of geometry of the emerging field theory? The derivation is not constructive enough to allow us to understand this point. »

The external potential fixes the density profile. Since the velocity of gapless excitations is density-dependent, this directly affects the velocity $v(x)$. The velocity itself enters the action of the field theory through the metric. As the referee rightly points out, our claim is that the position-dependent metric is sufficient to grasp the effects of the external potential V(x). Our derivation is constructive in the sense that we argue that, in order to fix the underlying field theory action, we only need to know the behavior of correlation functions at the scale $\ell$ where the system is locally homogeneous.
We are flattered that the referee thinks this is non-trivial reasoning, as it is the main idea in the paper. It is supported by the perfect agreement between numerics and the exact formulas that we are able to derive in the paper. It can also be supported by exact calculations and asymptotic analysis of, say, the Hermite kernel in the case of a harmonic trap.

B. « I also think it might be good to discuss cases near criticality. What would happen in models where the correlation length is not infinite, and comparable or smaller than the inhomogeneity length scale? Do we get a massive (or otherwise nonconformal) model on a curved space? »

This is an interesting suggestion. Presumably, this would induce a mass term in the field theory action, as the referee mentions. This is beyond the scope of the present paper though, and we have not studied this situation yet; we will perhaps come to this in future work.

---

## Editorial Decision

published